# Photocatalytic CO₂ Reduction Coupled with Alcohol Oxidation over Porous Carbon Nitride

Chuntian Qiu [1,2,†], Shan Wang [1,†], Jiandong Zuo [1,*] and Bing Zhang [2,3,*]

[1] College of Materials Science and Engineering, Shenzhen University, Shenzhen 518060, China; qiuct@szu.edu.cn (C.Q.); wangshanwork@yeah.net (S.W.)

[2] International Collaborative Laboratory of 2D Materials for Optoelectronics Science, Institute of Microscale Optoelectronics, Technology of Ministry of Education, Shenzhen University, Shenzhen 518060, China

[3] ZJU-Hangzhou Global Scientific and Technological Innovation Center, Zhejiang University, Hangzhou 311200, China

\* Correspondence: jdzuo@szu.edu.cn (J.Z.); bzhang219@zju.edu.cn (B.Z.)

† These authors contributed equally to this work.

**Abstract:** The photocatalytic transformation of CO₂ to valuable man-made feedstocks is a promising method for balancing the carbon cycle; however, it is often hampered by the consumption of extra hole scavengers. Here, a synergistic redox system using photogenerated electron-hole pairs was constructed by employing a porous carbon nitride with many cyanide groups as a metal-free photocatalyst. Selective CO₂ reduction to CO using photogenerated electrons was achieved under mild conditions; simultaneously, various alcohols were effectively oxidized to value-added aldehydes using holes. The results showed that thermal calcination process using ammonium sulfate as porogen contributes to the construction of a porous structure. As-obtained cyanide groups can facilitate charge carrier separation and promote moderate CO₂ adsorption. Electron-donating groups in alcohols could enhance the activity via a faster hydrogen-donating process. This concerted photocatalytic system that synergistically utilizes electron-hole pairs upon light excitation contributes to the construction of cost-effective and multifunctional photocatalytic systems for selective CO₂ reduction and artificial photosynthesis.

**Keywords:** concerted catalysis; CO₂ reduction; alcohol oxidation; photocatalysis; metal-free catalyst

## 1. Introduction

The capture and conversion of excess anthropogenic CO₂ are of great significance for repairing the current natural carbon cycle and relieving the global energy and environmental crisis [1–3]. However, due to chemical inertness and extremely high bond energies of the C=O bond (806 kJ/mol), the effective activation and reutilization of CO₂ under mild conditions are great challenges but are also of utmost importance for practical applications [3–5]. Photochemical reduction via artificial photosynthesis is a promising method to convert solar to chemical energy in the form of CO₂-fixation feedstocks under mild conditions [6–8]. Since pioneering work in the 1980s, great varieties of photoinduced CO₂ capture and/or transformation systems involving efficient light-absorbing semiconductors, such as CN, TiO₂, and CuIn₅S₈ [9–11], and electron transfer mediators, such as metals and/or metal−ligand complexes [12–14], have been developed. However, with respect to effectiveness and scale-up considerations, cost-effective and specifically nonnoble metal photocatalysts are highly desired.

Carbon nitride (CN), an inexpensive, metal-free semiconductor that absorbs visible light, is a promising photocatalyst candidate that enables the generation of high-energy electrons for H₂O and/or CO₂ reduction catalysis [15–18]. Moreover, the tris-triazine-based covalent framework in CN contains abundant surface sites with different basicities, such

as cyanide groups and amido groups, which can facilitate the capture and activation of thermodynamically stable $CO_2$ to different extents. Various products, such as CO, $CH_4$, and HCOOH, can be obtained by tuning the basic surface groups or deposited metal species [19]. CNs have been widely employed for photochemical $CO_2$ fixation with the assistance of hole scavengers [20–22], which are used to accelerate the slow oxidation half-reaction and to promote the separation of photogenerated charges. However, the consumption of hole scavengers increases costs and decreases atomic economy. Alternative systems that fully utilize photogenerated electron–hole pairs for the manufacture of $CO_2$-fixation feedstocks and value-added oxidized chemicals are more attractive from the viewpoint of green chemistry [23–25].

Recently, coupled photocatalysis systems that synergistically utilize electrons and holes for the controllable synthesis of value hydrogenated and oxidized chemicals, such as N-alkyl drugs, have been reported [26–28]. To fulfill these goals, photocatalysts with fine structure are vital for reaching high efficiency. Huang et al. reported a facile process to fabricate sulfur-doped carbon nitride to facilitate the separation of photogenerated electrons and holes [29]. Due to severe environmental issues and increasing concerns on carbon neutralization, the conversion of $CO_2$ into value-added chemicals on various advanced catalysts has drawn more attentions [30,31]. The abundant nitrogen atoms within the framework of carbon nitride also help in constructing single metal sites and converse $CO_2$ into deep reductive products [32]. Based on these achievements, a porous CN with abundant cyanide groups was synthesized via thermal calcination processes using ammonium sulfate as a porogen, and the as-obtained porous carbon nitride can facilitate $CO_2$ adsorption and charge separation. An effective concerted photocatalytic system was constructed for selective $CO_2$ reduction to CO in the gas phase, and a series of alcohols was simultaneously oxidized to value-added aldehydes and ketones in the liquid phase. The results provide more insight into the relationship between structure and photocatalytic performance and contribute to the design of cost-effective multifunctional photocatalysts and concerted green photocatalytic systems for value-added fine chemical production.

## 2. Results and Discussions

XRD was first performed to determine the structure of the synthesized CNs. As shown in Figure 1a, representative diffraction peaks of the (002) plane centered at approximately 27.5° were observed; the peak at ~13° for carbon nitride originated from a planar ordering parallel to the c-axis, which also illustrates the nature of carbon nitride. These results suggested that the synthesized samples had a typical carbon nitride structure. [15,24] With the increase in porogens (($NH_4$)$_2SO_4$), tiny diffraction peaks emerged at approximately 21.8°; these peaks may be a result of the minor reconfiguration of structural units via the incomplete thermal condensation process [24,33,34]. In addition to the bulk structure, porogen also resulted in the rearrangements of surface functional groups, which were also observed in the FT-IR spectra (Figure 1b). The typical vibration peaks at approximately 800 cm$^{-1}$ were assigned to the triazine/heptazine rings, and the strong adsorption at 3000–3500 cm$^{-1}$ can be assigned to stretching mode of N-H groups, while the peaks between 1200 and 1700 cm$^{-1}$ were assigned to the vibrations of CN heterocycles [33–35]. Remarkably, enhanced response peaks assigned to the cyano group (−C≡N) at approximately 2180 cm$^{-1}$ [24,33,34] were observed in the CN-S samples instead of the bulk CN. This result indicates that the decomposition of porogen in the pore-forming process may have led to incomplete condensation; thus, more terminal groups were formed instead of highly polymerized heterocycles and/or long chains. Furthermore, porogen contributes to the construction of porous structures that benefit substrate adsorption/desorption and mass transfer. With the increase in porogens, a higher BET surface area of CN-S samples of more than 22 m$^2$·g$^{-1}$ was obtained compared with that of bulk CN (9.8 m$^2$·g$^{-1}$) (Figure 1c). All these samples showed typical mesoporous characteristics in the $N_2$ adsorption–desorption isotherms, suggesting the absence of inherent microporosity. The majority of mesopores were in a narrow range of 2~3 nm (Figure 1d). In addition, a broad distribution from

5 to 50 nm, which was attributed to the stacking holes, was observed, while the number of mesopores gradually increased with increasing porogen content. Nevertheless, excess porogen blocked the creation of mesopores (Figure 1d, CN-S6). Thus, an appropriate amount of porogen could lead to the reconfiguration of the surface groups and improve both structural and textural properties that might enhance catalytic activities.

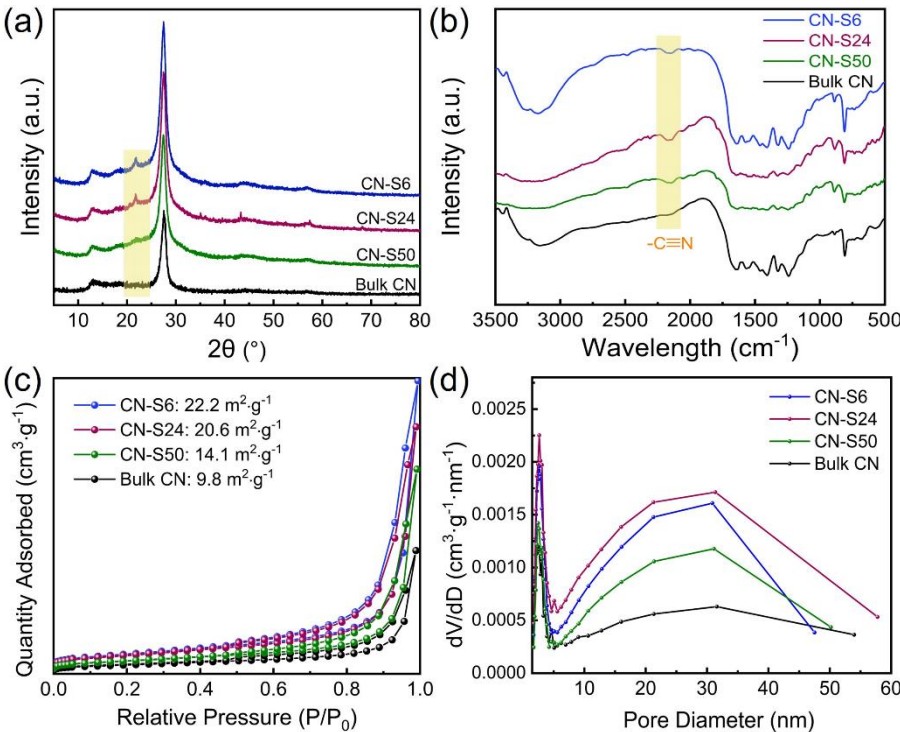

**Figure 1.** Structural characterizations of as-synthesized bulk CN, CN-S50, CN-S24, and CN-S6 samples: (**a**) XRD patterns; (**b**) FT-IR spectra; (**c**) $N_2$ adsorption–desorption isotherms; (**d**) pore size distributions.

Subsequently, the morphologies and compositions of the synthesized CNs were characterized by TEM and EDS. In contrast to that of the bulk CN (Figure 2a), a mesoporous structure was obtained via thermal polymerization with the assistance of ammonia sulfate porogen (Figure 2b); this observation coincides with the $N_2$ adsorption–desorption results. Significantly, O and trace S were detected by both EDS and XPS (Figures S1 and S2), suggesting that S is not incorporated into the structure, which is quite different with results reported in previous literature [29]. Gaseous sulfur-containing compounds may be generated via the high-temperature calcination process and result in structural porosity and a larger surface area that are favorable for catalysis.

Next, the band structure of the synthesized CNs was determined by UV–vis DRS and Mott–Schottky plots (Figure 3). The pore-making process will hinder polymerization to form CN heterocycles and follow the triazine/heptazine rings, which respond to visible light absorption. Thus, light absorption gradually decreased with increases in porogen, and the bandgaps of those CN-S samples were slightly broadened. As shown in Figure 3c, the bandgaps of all samples were sufficient for water splitting, alcohol oxidation, and $CO_2$ reduction reactions. $CO_2$ reduction to $CH_4$ is more thermodynamically favorable than CO formation; however, the formation of $CH_4$ involves an eight-electron transfer process, which is much more kinetically difficult than two-electron reduction to CO. To accelerate photoinduced electron separation and transfer, photogenerated holes paired with electrons are usually quenched by excess highly reductive scavengers, such as methanol and triethanolamine [20–22]. In this manner, the photocatalytic activities of the reduction half-reaction could be greatly enhanced; however, the consumption of the hole scavengers increases the cost and limits the atomic economy. In this regard, the synthesized catalysts

are capable of being employed in synergistic photocatalytic systems and are expected to fully utilize the photogenerated electron–hole pairs to generate value-added oxidation products and simultaneously enhance $CO_2$ reduction to desired $C_1$ fuels such as CO, $CH_3OH$, and $CH_4$.

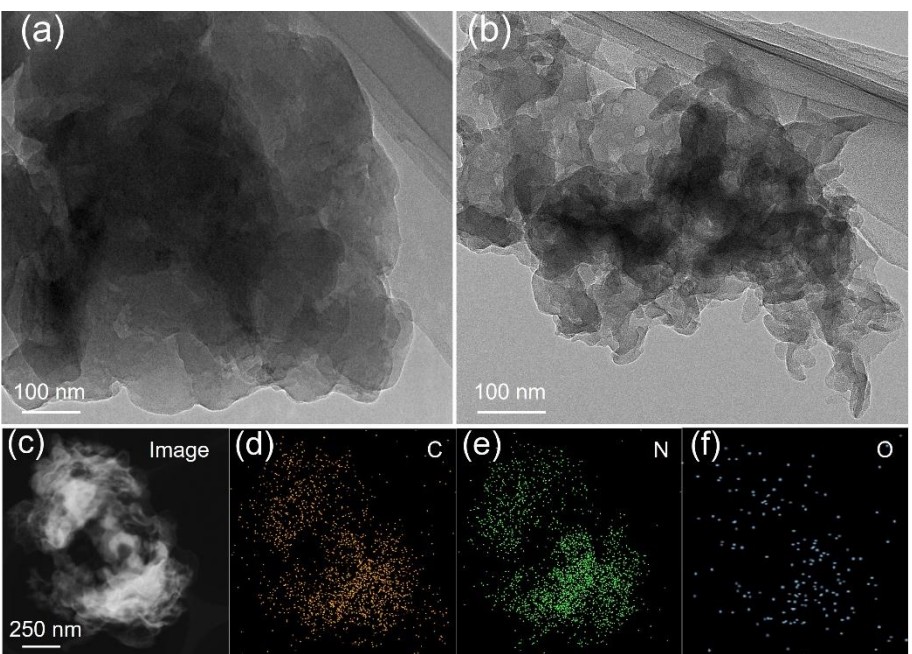

**Figure 2.** TEM images of the synthesized CN (**a**) and CN-S24 (**b**) samples; EDS mapping of CN-S24 (**c**–**f**).

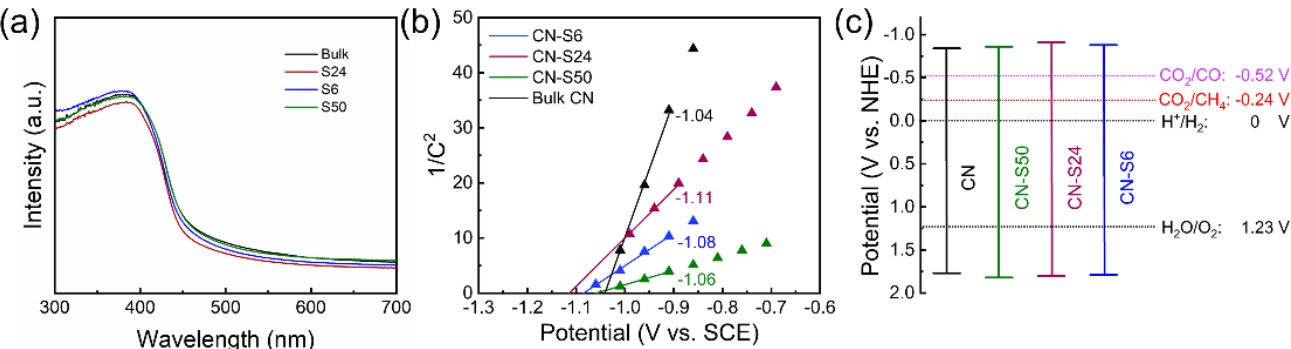

**Figure 3.** Bandgap characterization of the synthesized CN, CN-S50, CN-S24, and CN-S6 samples: (**a**) UV–vis diffuse reflectance spectra. (**b**) Mott–Schottky plots in a 0.5 M $Na_2SO_4$ aqueous solution. (**c**) Calculated bandgaps.

To determine the best photocatalyst candidate and optimize the reaction's conditions, alcohol oxidation was first conducted over the synthesized CN-S samples under an $O_2$ atmosphere, and the yields were determined by GC-MS characterization (Table S1). An approximately 32% yield of benzaldehyde was obtained over bulk CN within 4 h, while porous CN-S24 achieved the highest yield of up to 93% when the reaction time was prolonged to 8 h in MeCN. Remarkably, an approximately 65% yield was obtained under an Ar atmosphere, demonstrating that the photogenerated holes are capable of oxidizing aromatic alcohols and releasing hydrogen without the assistance of $O_2$. In addition, functional groups on the benzene ring noticeably impacted photocatalytic activities. Electron-withdrawing groups such as nitro and halogen groups reduced activities, while electron-donating groups such as methoxy groups enhanced activities (Figure 4a) [24]. These results enabled the construction of a green concerted photocatalytic system that utilizes photogenerated holes for the oxidation of alcohols to value-added aldehydes and/or ketones and furnishes H atoms that could be involved with photoexcited

electrons for the production of $CO_2$-fixation feedstocks (CO (Figure 4b)). CN-S24 yielded much more benzaldehyde (~9.1 $\mu mol \cdot h^{-1}$) than bulk CN. When p-methoxybenzyl alcohol was used as the substrate, the yield of p-methoxybenzaldehyde increased to approximately twice that of bulk CN (~11.3 $\mu mol \cdot h^{-1}$). Simultaneously, hydrogen furnished by alcohol oxidation was transferred to reduce the $CO_2$ gas with the assistance of photogenerated electrons. Approximately 1.8 $\mu mol \cdot h^{-1} \cdot g^{-1}$ CO was obtained over bulk CN, which is much lower than that over CN-S24 (6.5 $\mu mol \cdot h^{-1} \cdot g^{-1}$). The performance of the CN-S24 sample for photocatalytic $CO_2$ reduction surpassed that of most carbon nitride photocatalysts and was even comparable to that of metal-loaded carbon nitride catalysts (Table S2). In addition, the obtained CN-S24 sample exhibited good reusability over three cycles (Figure S3) and maintained a typical carbon nitride structure, as indicated by the XRD patterns (Figure S4). These results demonstrated that CN-S24 is a robust and efficient photocatalyst that can achieve the green utilization of photogenerated electron–hole pairs for synergistic alcohol oxidation and $CO_2$ fixation.

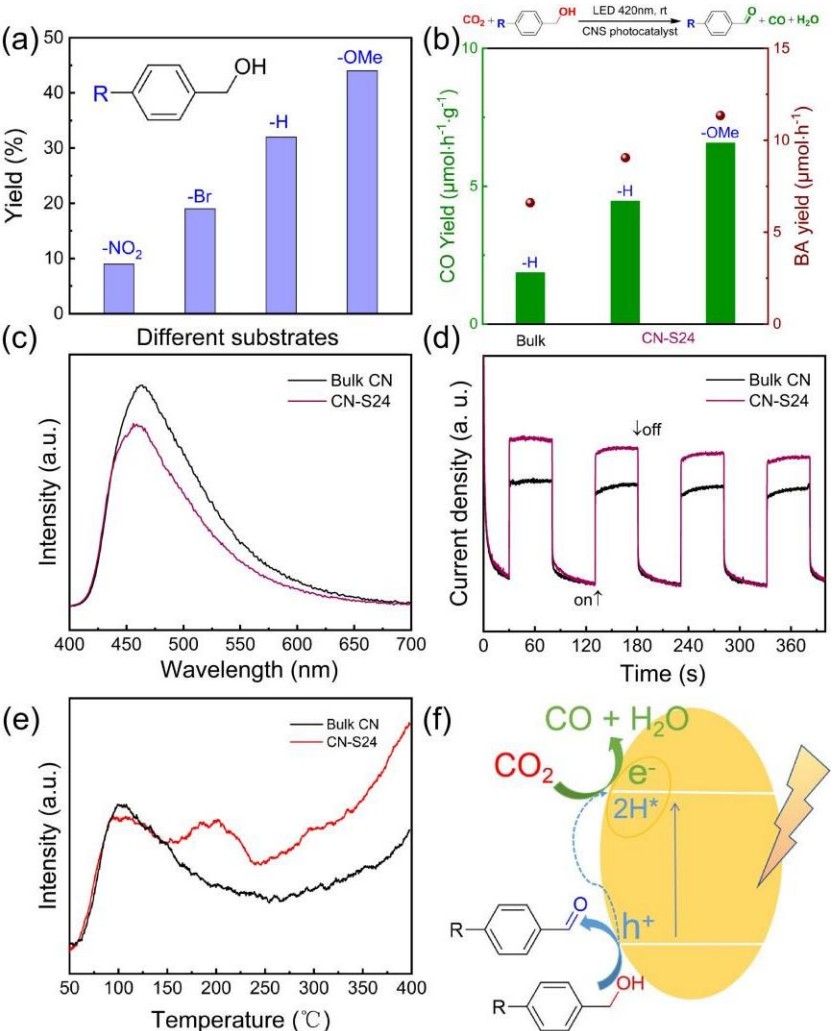

**Figure 4.** (**a**) Substrates with different substituent groups involved in the oxidation half-reaction over CN-S24. Reaction conditions: 0.1 mmol of substrates, 5 mL of $CH_3CN$, 20 mg of catalyst, room temperature, 4 h, and Ar atmosphere. (**b**) Simultaneous CO production and alcohol oxidation over the synthesized CN-S24 and control photocatalysts. Reaction conditions: 0.1 mmol of substrates, 5 mL of $CH_3CN$, 20 mg of catalyst, room temperature, 2 h, and $CO_2$ atmosphere. (**c**) PL spectra of bulk CN and CN-S24. (**d**) On/off photocurrent response of bulk-CN and CN-S24 in 0.5 M $Na_2SO_4$ solution. (**e**) $CO_2$-TPD of bulk-CN and CN-S24. (**f**) Proposed reaction pathway of $CO_2$ reduction to CO and alcohol oxidation on the photocatalyst.

Subsequently, photoluminescence spectroscopy (PL), photocurrent response, and $CO_2$-TPD experiments were conducted. CN-S24 exhibited a lower fluorescence intensity than bulk CN, indicating more efficient charge separation (Figure 4c), which further strengthened the photocurrent of CN-S24 (Figure 4d) [24–27,36]. These enhanced photoelectronic properties may be due to the reconfiguration of surface functional groups and an increase in cyanide groups. It has been reported that the strong electron withdrawing capacity of $-C\equiv N$ groups contributes to the delocalization of the isolated valence electrons in the π-conjugated heterocyclic rings and, thus, increases the concentration of delocalized electrons, which could facilitate the photocatalytic generation of active radical species [24,33,34]. In addition, enhanced $CO_2$ adsorption was observed in the temperature region beyond 150 °C (Figure 4e). This result indicates that more chemical adsorption sites with moderate $CO_2$ adsorption capacity may emerge in CN-S24, which may facilitate $CO_2$ adsorption and activation for the subsequent transformation of CO [37]. Thus, due to those enhanced structural and photoelectronic properties, CN-S24 achieved much better performance toward the utilization of photogenerated electron–hole pairs for synergistic alcohol oxidation and $CO_2$ reduction to CO via the proposed reaction path (Figure 4f).

## 3. Materials and Methods

### 3.1. Preparation of Carbon Nitride with Porogens

Cyanamide (Aladdin Biochemical Technology Co., Ltd., Shanghai, China, 98 %), ammonium sulfate (Adamas Reagent Co., Ltd, Shanghai, China, 99%), acetonitrile (HPLC, Adamas Reagent Co., Ltd, Shanghai, China, 99.9%), and benzyl alcohol (Adamas Reagent Co., Ltd, Shanghai, China, 99%) were used as received.

First, 1 g of cyanamide and 0.13 g of ammonium sulfate (mol ratio 24:1) were dissolved in 3 mL of deionized water under sonication for 10 min. The solution was then transferred to a crucible with a cover. After heating at 550 °C for 4 h with a ramp rate of 2.3 °C min$^{-1}$ in air, light-yellow carbon nitride was obtained (0.58 g in total); this product is referred to as CN-S24. CN-6 and CN-50 samples were prepared following the above procedures, with the exception that ammonium sulfate was used at 0.52 g and 0.062 g, respectively. A bulk carbon nitride sample was prepared with pure cyanamide.

### 3.2. Photocatalytic Reactions

For the photocatalytic oxidation of alcohols, typically, 20 mg of catalyst, 0.1 mmol of alcohol, and 5 mL of solvent were mixed in a sealed 30 mL quartz bottle and saturated with oxygen or Ar gas. Then, the suspensions were irradiated under a 15 W LED lamp (420 nm). The products were analyzed with a GC–MS System (Agilent Technologies, G7036A, Santa Clara, CA, USA) using toluene as an external standard. For the photocatalytic reduction of $CO_2$, 20 mg of catalyst, alcohol (0.1 mmol) and 5 mL solvent were mixed in a 30 mL sealed quartz bottle and saturated with $CO_2$. Then, the suspensions were irradiated under a 15 W LED lamp (420 nm) for 2 h. The products were analyzed with a GC System (Fuli instrument, GC9720).

### 3.3. Electrochemical and Photoelectrochemical Measurements

A conventional three-electrode cell system connected to a CHI 660E (Chenhua, Shanghai, China) electrochemical workstation was applied for further evaluation. Samples dropped on an ITO slide served as the working electrode, and a saturated calomel electrode (SCE) and a Pt wire were used as the reference electrode and counter electrode, respectively. Mott–Schottky plots and on/off photocurrent responses were obtained in 0.5 M $Na_2SO_4$ aqueous solution. A 300 W Xe lamp (PLS-SXE300/300UV, Perfect Light, China) equipped with a 420 nm cutoff filter was used for photocurrent detection.

### 3.4. Characterization

Scanning electron microscopy (SEM) measurements were performed on a MIRA3 TESCAN scanning electron microscope. Transmission electron microscopy (TEM) measure-

ments were performed with a HITACH HT7700 microscope operated at an acceleration voltage of 80 kV. X-ray diffraction (XRD) (Ultima IV, Rigaku) was performed at 40 kV and 40 mA (Cu K$\alpha$ X-ray radiation source) with a scanning speed of $6°$ min$^{-1}$. X-ray photoelectron spectroscopy (XPS) measurements were conducted on a Thermo Scientific K-Alpha XPS system using monochromate Al K$\alpha$ radiation. Nitrogen sorption experiments (BET) were performed on a Micromeritics ASAP 2460 (samples were degassed at 250 °C for 12 h before measurements). EPR (electron paramagnetic resonance) tests were conducted on a BRUKER A300-10/12 instrument. IR (infrared spectroscopy) was performed on a Nicolet 6700. UV–vis spectra were taken with an Agilent Cary Series UV–vis-NIR spectrophotometer.

## 4. Conclusions

Porous carbon nitrides with substantial cyanide groups were synthesized and employed as metal-free photocatalysts for simultaneous alcohol oxidation and $CO_2$ reduction to CO by the synergistic utilization of photogenerated electron–hole pairs. This method of preparing cheap but efficient carbon nitride photocatalysts is facile and easy to produce in large scale, and the optimized CN-S24 showed the best performance mainly due to the improved structural properties and exposed cyanide groups that helped facilitate charge separation and $CO_2$ adsorption. It was found that alcohols with electron-donating groups could enhance activities via a faster hydrogen-donating processes, and an appropriate CO production rate of 6.5 $\mu mol \cdot h^{-1} \cdot g^{-1}$ was achieved. These results contributed to the fabrication of highly efficient photocatalysts and the construction of concerted photocatalytic systems that fully utilize electron-hole pairs. The developed synergistic system for photochemically producing value-added chemicals as well as for reducing $CO_2$ molecules can facilitate the kinetic process, and it can be applied in other systems such as water splitting and organic synthesis.

**Supplementary Materials:** The following supporting information can be downloaded at: https://www.mdpi.com/article/10.3390/catal12060672/s1, Figure S1: EDX of as-synthesized CN-S24 sample; Figure S2: XPS survey spectra of as-synthesized CN, CN-S50, CN-S24, and CN-S6 samples; Figure S3: Photocatalytic CO yields on CN-S24 sample for different cycles; Figure S4: XRD pattern of CN-S24 sample after reaction; Table S1: Photocatalytic activities over as-synthesized CNs samples for alcohol oxidation; Table S2: Photocatalytic activities over reported carbon nitride-based samples for $CO_2$ reduction. References [38–47] are citied in the Supplementary Materials.

**Author Contributions:** Conceptualization, J.Z. and B.Z.; methodology, C.Q. and S.W.; validation, B.Z; formal analysis, C.Q and J.Z.; investigation, S.W.; resources, S.W; data curation, C.Q.; writing—original draft preparation, C.Q. and S.W.; writing—review and editing, J. Z. and B. Z.; supervision, B.Z.; project administration, B.Z.; funding acquisition, C.Q. and B.Z. All authors have read and agreed to the published version of the manuscript.

**Funding:** This work was financially supported by the National Natural Science Foundation of China (21902105), Guangdong Basic and Applied Basic Research Foundation (2020A1515010471), Shenzhen Science and Technology Program (JCYJ20210324094000001), and Chongqing Key Laboratory for Advanced Materials & Technologies of Clean Energies (JJNY202003).

**Data Availability Statement:** The data presented in this study are available on request from the corresponding author.

**Acknowledgments:** The authors wish to acknowledge the assistance on TEM received from the Electron Microscope Center of the Shenzhen University.

**Conflicts of Interest:** The authors declare no conflict of interest.

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
