# Peer review of "Photocatalytic CO2 Reduction Coupled with Alcohol Oxidation over Porous Carbon Nitride"

_catalysts, doi:10.3390/catal12060672_

Round 1
Reviewer 1 Report
The authors presented simultaneous CO2/CO reduction and alcohol oxidation over porous carbon nitride prepared from cyanamide and ammonium sulfate. The work is sufficient for publication and it will be interesting to the readers and researchers for CN materials and CO2 valorization. Minor comments can be addressed.
1- please interpret the peak at ~12.8 degree in the XRD (Fig. 1A) and dd it to the discussion.
2- FTIR fig 1b, please comment on the strong absorption at 3000-3500 cm(-1) in bulk and S6 CN samples? N-H groups?
3- Did the authors perform mass measurement for the samples (MALDI TOF), if the authors can do or extract some information from mass can help to understand the structures better and the active species in the porous-CN materials?
4- in graph 4b, please align the colours of the bars (CO) with the yaxis to help readers read the results.
5- the reaction yields were improved with longer reaction times (from 4h to 8h as shown in the table in the SI). I am wondering why the authors chose 4 h to study the simultaneous reaction? Did the authors study CO evolution over reaction time,if so, please present such data in an additional figure in the manuscript. If not, please comment on the reaction time.
6- Similar work prepared material from dicyandiamide and sulfates (phys chem Chem Phys 2020 May 14;22(18):10116-10122. doi: 10.1039/c9cp07002h.
Sulfate modified g-C 3 N 4 with enhanced photocatalytic activity towards hydrogen evolution: the role of sulfate in photocatalysis)
please add it to the introduction and cite it, compare the materials obtained to this work.
7-I advise to move Table S2 from SI to the introduction if it was not published in a recent review and if it is original comparison of the authors. I find that it will be useful and interesting to readers.
8- please cite the following more more recent references on CO2 valorization especially that this work targets simultaneous reactions, so readers can combine the state of the art. the first cited reference in your manuscript is 2014, and this field of research is very fast.
Carbon Dioxide Capture and Conversion-Advanced Materials and Processes - Elsevier 1st Edition - June 1, 2022- ISBN: 9780323855853
and
Introductory Chapter: An Outline of Carbon Dioxide Chemistry, Uses and Technology-2018-DOI: 10.5772/intechopen.79461
Reviewer 2 Report
In the article "Photocatalytic CO2 Reduction coupled with Alcohol Oxidation over porous carbon nitride" the authors presented their research. The work consists of four parts: introduction, materials and methods, results and discussions, and conclusions. The first part is devoted to the literature review. The next part is the materials and methods, in which the authors of the article present preparation of carbon nitride with porogens, photocatalytic reactions, electrochemical and photoelectrochemical measurements. The next part concerns the results obtained by the authors - structural analysis. The obtained studies were properly described and interpreted. All the results contained in it have been presented in a very clear and legible way for the recipient. The conclusions and summary of the study are consistent and well-worded. Summing up, the reviewed work presents a very high substantive and experimental value. Comments for authors:
1) too little information about the research in the abstract,
2) too short a description of the literature review of the described issue,
3) The added value of the reviewed work would be the inclusion of GC-MS spectra with reaction products for the tested systems,
4) the conclusions should supplement / emphasize the advantages of the proposed method.
